# Automatic Contrast Phase Detection on Abdominal Computed Tomography using Clinically-Inspired Techniques

**Eduardo Pontes Reis**[1,2]                                    EDREIS@STANFORD.EDU
[1] *Stanford University, CA, USA*
[2] *Hospital Israelita Albert Einstein, Sao Paulo, Brazil*

**Louis Blankemeier**[1]                               LOUIS.BLANKEMEIER@STANFORD.EDU
**Juan Manuel Zambrano Chaves**[1]                               JMZ@STANFORD.EDU
**Malte Engmann Kjeldskov Jensen**[1]                            MEKJ@STANFORD.EDU
**Sally Yao**[1]                                           YAOHANQI@STANFORD.EDU
**Cesar Augusto Madid Truyts**[2]                       CESAR.TRUYTS@EINSTEIN.BR
**Marc Willis** [1]                                     MARC.WILLIS@STANFORD.EDU
**Robert Downey Boutin**[1]                                BOUTIN@STANFORD.EDU
**Edson Amaro Jr**[2]                                  EDSON.JUNIOR@EINSTEIN.BR
**Akshay Chaudhari**[1]                                  AKSHAYSC@STANFORD.EDU

**Editors:** Under Review for MIDL 2023

## Abstract

Accurately determining contrast phase in an abdominal computed tomography (CT) series is an important step prior to deploying downstream artificial intelligence methods trained to operate on the specific series. Inspired by how radiologists assess contrast phase status, this paper presents a simple approach to automatically detect the contrast phase. This method combines features extracted from the segmentation of key anatomical structures with a gradient boosting classifier for this task. The algorithm demonstrates high accuracy in categorizing the images into non-contrast (96.6% F1 score), arterial (78.9% F1 score), venous (92.2% F1 score), and delayed phases (95.0% F1 score), making it a valuable tool for enhancing AI applicability in medical imaging.

**Keywords:** Abdominal CT Scan, Contrast Phase, Organ Segmentation, Radiology, Medical Imaging, Machine Learning, Artificial Intelligence

## 1. Introduction

Abdominal computed tomography (CT) scans are commonly utilized to assess internal organs and structures. CT exams can be performed by scanning subjects in different conditions (phases) related to the use of intravascular contrast agents, which enhance the radiodensity of blood vessels and vascularized internal organs. Accurate determination of phase is crucial, especially as the quantification of biomarkers in the rapidly emerging field of opportunistic imaging relies on it. (Pickhardt et al., 2013; Zambrano Chaves et al., 2021) This ensures that the appropriate algorithm runs on the correct series of images and that quantitative metrics are calibrated for phase status. (Boutin et al., 2016)

To the best of our knowledge, current methods for contrast phase detection in abdominal CT scans are not available through open-source platforms. (Dao et al., 2022; Ye et al., 2022)

In this scenario, projects that analyze CT images are required to manually curate extensive datasets for classifier training, which can be both costly and time-consuming.

In order to address these limitations, we present the Contrast Phase algorithm, which extracts radiodensity measures from relevant organs and applies a gradient boosting classifier for accurate contrast phase classification. The algorithm identifies four contrast phases: non-contrast, arterial, venous, and delayed. This pipeline is design to read most common image formats (DICOM and NIFTI), segment relevant organs, and classify contrast phases. It is made publicly available at https://github.com/StanfordMIMI/Comp2Comp

## 2. Methods

All data aggregation and experiments were performed under Institutional Review Board approval using de-identified clinical data.

The data acquisition and labeling process for this study involved obtaining 739 abdominal CT exams from 238 unique patients. These CT exams contained 3252 series. Sagittal and coronal reformatted series, localizer series, and axial series that failed during organ segmentation were excluded. 1545 remaining axial series were split into the training set, containing 1183 examples, and the test set, containing 362 examples. We ensured that each patient's data was exclusively allocated to either the training or the test set.

The series were then labeled as one of 4 classes: `non-contrast`, `arterial`, `venous` and `delayed`. To facilitate the labeling process of contrast phases in the CT scans, the Series Description DICOM tag was analyzed using regular expression (regex) rules to detect specific keywords describing the phases along with their synonyms. This initial labeling process served as a preliminary categorization of the scans. Subsequently, a board-certified radiologist (5 years of experience) reviewed two slices from each scan, on the anatomical levels of the right adrenal gland and the left kidney, where the structures to evaluate contrast phase are most discernible. Any inaccurately labeled scans were corrected.

The Contrast Phase algorithm consists of three main stages: segmentation of organs, feature extraction, and classification.

**Segmentation of Organs:** The first stage involves the segmentation of key anatomical structures, including the aorta, inferior vena cava, portal vein, renal parenchyma, and renal pelvis, using Total Segmentator, a deep learning-based open-source segmentation tool (Wasserthal et al., 2022). This approach has been shown to provide accurate and precise organ segmentation, which is crucial for the subsequent feature extraction of these regions.

**Feature Extraction:** After segmentation, we computed quantitative 48 low-level radiomics features that characterize the radiointensity statistics, such as maximum value, minimum value, mean, median, and variance from the aforementioned anatomical structures.

**Classification:** Extreme Gradient Boosting (XGBoost) (Chen and Guestrin, 2016) was trained on the extracted features to classify the CT images into the four distinct contrast phases.

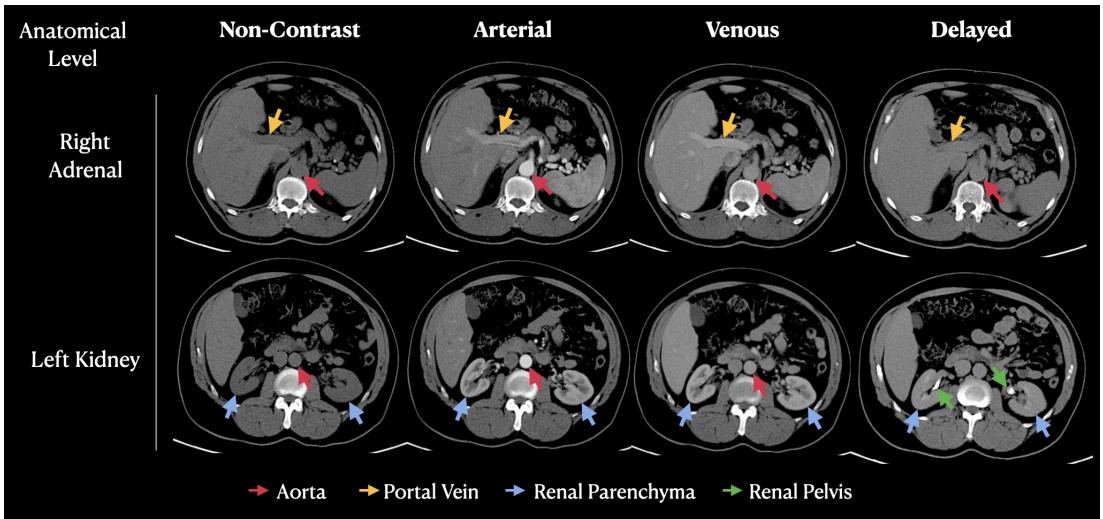

Figure 1: Example abdominal CT images at the anatomical level of the right adrenal gland and the left kidney, representing each of the four classes. Observe the variations in pixel intensity across the phases in the key structures: aorta (red arrows), portal vein (yellow arrows), renal parenchyma (blue arrows), and renal pelvis (green arrows).

## 3. Results and Discussion

The classifier demonstrated high accuracy on the test set in identifying the four contrast phases, with an accuracy of 92.3% and F1-scores of 96.6% for non-contrast, 78.9% for arterial, 92.2% for venous, and 95.0% for delayed phase, as shown in Table 1. These results highlight that using segmentations of clinically relevant anatomic structures can contribute to the development of an accurate contrast phase classifier. The arterial phase was the most challenging to classify, which could be attributed to the limited number of training examples for this phase. This model can serve as a valuable component in a pipeline of other AI algorithms for abdominal CT scan analysis.

| Metric | Non-Contrast | Arterial | Venous | Delayed |
|---|---|---|---|---|
| # Training examples | 285 (24.0%) | 49 (4.1%) | 503 (42.5%) | 346 (29.2%) |
| # Test examples | 76 (20.9%) | 22 (6.0%) | 139 (38.4%) | 125 (34.5%) |
| Precision | 100.0 | 93.7 | 87.1 | 97.4 |
| Recall | 93.4 | 68.1 | 97.8 | 92.8 |
| Specificity | 100.0 | 99.7 | 91.0 | 91.0 |
| F1 score | 96.6 | 78.9 | 92.2 | 95.0 |

Table 1: Performance metrics and dataset distribution for the classification model

The algorithm has been made publicly available through the Comp2Comp Inference Pipeline - Open-Source Body Composition Assessment on Computed Tomography (Blankemeier et al., 2023) on the following GitHub repository: https://github.com/StanfordMIMI/Comp2Comp. We provide an easy-to-use command line interface that operates on DICOM and NIfTI medical image formats.

## 4. Conclusion

We introduce an efficient algorithm for detecting contrast phases in abdominal CT scans. We show that by carefully choosing key structures to extract features, we achieve high accuracy for contrast phase classification. While the current focus is on the abdominal region, this method has the potential to be expanded to additional fields of view.

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
