# OpenReview forum: "Automatic Contrast Phase Detection on Abdominal Computed Tomography using Clinically-Inspired Techniques"
_MIDL.io/2023/Short_Paper_Track — MIDL 2023 Short paper track Poster_

### Official Review · Reviewer_kpzF · 2023-04-19
**Great article on detecting contrast phase in abdominal CT**

**Rating:** 9
**Confidence:** 5

**Review:**

By itself Computed Tomography imaging provides poor contrast of vascular structures. Contrast agent has to be injected in the bloodstream to make such structures visible. Depending on when the images are acquired the images can capture the opaque blood while in the arteries, in the veins or in the organs. Determining what "phase" an image is at is important for downstream tasks.

This article presents a simple and effective method for automatically detecting the contrast phase in abdominal computed tomography (CT) scans. The proposed method combines features extracted from the segmentation of key anatomical structures and a gradient boosting classifier.

Validation is proposed on N=739 CT exams.

Implementation is made publicly available as part of a larger software package here:
https://github.com/StanfordMIMI/Comp2Comp

Pros:
* a very clear paper!
* a clear focus on practicality by being inspired by how radiologists assess contrast phase status.
* a no-nonsense efficient gradient boosting classifier that works great for the task.
* there is potential for expanding this method to additional fields of view.
* large validation dataset
* public implementation that appears well documented and very usable

Cons:
* lack of comparison/becnhmarking with other existing methods for contrast phase detection (which to the author credit might not be available...)

Ideas for Related and Future Work:
* Increase the number of training examples, particularly for the arterial phase, to improve the performance of the classifier.
* If possible, compare the proposed method with existing approaches for contrast phase detection to demonstrate its advantages and potential limitations.
* Explore the possibility of fine-tuning the feature extraction process to further enhance the algorithm's performance.
* Investigate the method's applicability in other medical imaging modalities, such as magnetic resonance imaging (MRI) or positron emission tomography (PET) scans.
* When included in complete pipeline, asses benefit to clinical decision making and patient outcome?

---

### Official Review · Reviewer_A1oF · 2023-04-20
**Clear method, clinical value unclear**

**Rating:** 5
**Confidence:** 4

**Review:**

This short paper described a machine learning-based tool – the Contrast Phase algorithm – that can be used to determine the contrast phase of an abdominal CT scan automatically. The tool performs automatic segmentation and, subsequently, classification of scans based on features extracted in segmented organs. The method is straightforward and gives good results. Evaluation of the results is somewhat suboptimal and it does not become clear what the algorithm adds over labeling extracted from DICOM tags.

Strengths

-	The short paper addresses a potentially relevant problem in abdominal CT image analysis: different phases exist within many CT studies and based on DICOM tags alone it’s not always trivial to distinguish these.
-	The proposed method is provided as open source on GitHub and will likely be a useful tool.

Weaknesses

-	It would be interesting to report in how many cases the human observer had to correct the initial labeling obtained from DICOM tags in the annotation process. This would give a good indication of the necessity of this automatic tool and the completeness and correctness of DICOM tags.
-	Authors use Total Segmentator to obtain segmentations of organs in all four contrast phases, but I can imagine that this model does not perform equally well for all phases. How does this affect subsequent steps?
-	Reporting of results could be clearer. If I understand correctly, the authors trained an Extreme Gradient Boosting classifier in a four-class set-up. However, the organization of Table 1 suggest that four individual binary models were trained. It would be more interesting to present a confusion matric/contingency table so that the reader can also which classes get confused with which other classes.